# The Bank Vole (*Clethrionomys glareolus*)—Small Animal Model for Hepacivirus Infection

**DOI:** 10.3390/v13122421

**Published:** 2021-12-03

**Authors:** Susanne Röhrs, Lineke Begeman, Beate K. Straub, Mariana Boadella, Dennis Hanke, Kerstin Wernike, Stephan Drewes, Bernd Hoffmann, Markus Keller, Jan Felix Drexler, Christian Drosten, Dirk Höper, Thijs Kuiken, Rainer G. Ulrich, Martin Beer

**Affiliations:** 1Institute of Diagnostic Virology, Friedrich-Loeffler-Institut, Federal Research Institute for Animal Health, 17493 Greifswald-Insel Riems, Germany; susanne.roehrs@googlemail.com (S.R.); dennis.hanke@fu-berlin.de (D.H.); kerstin.wernike@fli.de (K.W.); bernd.hoffmann@fli.de (B.H.); dirk.hoeper@fli.de (D.H.); 2Research Center for Emerging Infections and Zoonoses, University of Veterinary Medicine, 30559 Hannover, Germany; 3Department of Viroscience, Erasmus University Medical Centre, 3015 CN Rotterdam, The Netherlands; mariana.boadella@gmail.com (M.B.); t.kuiken@erasmusmc.nl (T.K.); 4Institute of Pathology, Johannes Gutenberg-University Mainz, 55131 Mainz, Germany; beate.straub@unimedizin-mainz.de; 5Centre for Infection Medicine, Department of Veterinary Medicine, Institute of Microbiology and Epizootics, Freie Universität Berlin, 14163 Berlin, Germany; 6Institute of Novel and Emerging Infectious Diseases, Friedrich-Loeffler-Institut, Federal Research Institute for Animal Health, 17493 Greifswald-Insel Riems, Germany; stephan.drewes@fli.de (S.D.); markus.keller@fli.de (M.K.); 7German Center for Infection Research (DZIF), Partner Site, Bonn-Cologne, 53113 Bonn, Germany; felix.drexler@charite.de (J.F.D.); christian.drosten@charite.de (C.D.); 8Institute of Virology, University of Bonn Medical Center, 53113 Bonn, Germany; 9Berlin Institute of Health, Institute of Virology, Charité-Universitätsmedizin Berlin, Corporate Member of Freie Universität Berlin, Humboldt-Universität zu Berlin, 10117 Berlin, Germany; 10German Center for Infection Research (DZIF), Partner Site, Hamburg-Lübeck-Borstel-Riems, 17493 Greifswald-Insel Riems, Germany

**Keywords:** animal model, virus–host interaction, rodent hepacivirus, bank vole, hepatitis C virus, *Hepacivirus F*, *Hepacivirus J*

## Abstract

Many people worldwide suffer from hepatitis C virus (HCV) infection, which is frequently persistent. The lack of efficient vaccines against HCV and the unavailability of or limited compliance with existing antiviral therapies is problematic for health care systems worldwide. Improved small animal models would support further hepacivirus research, including development of vaccines and novel antivirals. The recent discovery of several mammalian hepaciviruses may facilitate such research. In this study, we demonstrated that bank voles (*Clethrionomys glareolus*) were susceptible to bank vole-associated *Hepacivirus F* and *Hepacivirus J* strains, based on the detection of hepaciviral RNA in 52 of 55 experimentally inoculated voles. In contrast, interferon α/β receptor deficient C57/Bl6 mice were resistant to infection with both bank vole hepaciviruses (BvHVs). The highest viral genome loads in infected voles were detected in the liver, and viral RNA was visualized by in situ hybridization in hepatocytes, confirming a marked hepatotropism. Furthermore, liver lesions in infected voles resembled those of HCV infection in humans. In conclusion, infection with both BvHVs in their natural hosts shares striking similarities to HCV infection in humans and may represent promising small animal models for this important human disease.

## 1. Introduction

Hepatitis C virus (HCV), genus *Hepacivirus*, family *Flaviviridae*, has a high impact on global human health and economy. Currently, 115 (92–149) million people are acutely or chronically infected by this hepatotropic virus [1]. Chronic hepatitis C may progress to liver fibrosis and cirrhosis, and it increases the risk of developing hepatocellular carcinoma. Currently available antiviral drugs can be used to decrease viral replication to slow down the disease process, or even cure the disease [2]. Combined application of sofosbuvir, peginterferon and ribavirin for twelve weeks was described to clear the virus in patients with or without liver cirrhosis [3]. Although this represents a promising treatment, severe side effects are physically challenging. This negatively influences the compliance of the patients. Additionally, the combined treatment for 12 weeks costs up to EUR 100,000 per patient [4], and hence, both side effects and costs still restrict the treatment. Moreover, resistance-associated substitutions in the NS5A gene of HCV have been detected in a minority (5%) of treated patients [5]. Suitable small animal models are necessary, not only for development of efficacious and cost-effective vaccines [6] and antivirals, but also for detailed pathogenesis studies.

Chimpanzees (*Pan troglodytes*) and tree shrews (*Tupaia belangeri*) are thus far the only known species that develop similar clinical signs as humans upon experimental inoculation with HCV [7,8]. A non-human primate hepacivirus named GBV-B, which is distantly related to HCV, can cause hepatitis in experimentally infected New World monkeys and has been used in an alternative HCV infection model [9]. Ethical reasons and laboratory requirements exclude the general use of these species for studies on HCV. As hepaciviruses were recently detected in bats, horses, donkeys, cattle and dogs [10,11,12,13,14], these species may represent the basis for further animal models [15,16,17,18]. However, extensive experiments on these species are economically and ethically demanding. Furthermore, basic information on hepacivirus infections in dogs, donkeys and cattle is missing [11,14,17].

Models based on HCV-related Norway rat (*Rattus norvegicus*) hepacivirus infection or mouse-adapted HCV strains in mice, knock-out mice or mice transplanted with human hepatocytes can be inefficient or artificial and do not reflect a naturally occurring virus–host interaction, which is important to study the pathogenesis [8,10,19,20]. A small animal model based on a naturally occurring virus–host interaction, as has been achieved recently for Norway rat hepacivirus in Norway rats [21], would allow more detailed pathogenesis studies. This would greatly facilitate further research in the development of vaccines [22,23] and novel antivirals [21,24].

Since 2013, new hepacivirus sequences were identified in serum, liver and fecal samples of rodent species other than Norway rats, including bank voles (*Clethrionomys glareolus*, syn. *Myodes glareolus*), deer mice (*Peromyscus maniculatus*) and long-tailed ground squirrels (*Spermophilus undulatus*) [11,18,25,26,27]. Furthermore, real-time reverse transcription-polymerase chain reaction (RT-qPCR)-based positive strand analysis together with negative strand complementary RNA detection indicated a liver tropism and efficient replication for some of these hepaciviruses [25,26]. Additionally, a low-grade fibrosis was detected in bank vole hepacivirus (BvHV)-positive liver samples of naturally infected bank voles [25]. These results suggest similarities in the pathogenesis of BvHV-infection in their natural hosts and of HCV infection in humans.

In contrast to the other mammal species (such as horses or cattle), bank voles can be bred and maintained easily and economically under experimental conditions. In addition, bank voles were initially found to harbor two highly divergent BvHV clades [25], which could approximate the situation of HCV genotypes in humans. These BvHV lineages were recently classified as species *Hepacivirus F* (clade 2) and *Hepacivirus J* (clade 1) [28]. Both BvHV species were shown to have a broad geographical distribution in Europe with a sympatric occurrence at several sites in Germany [29]. Therefore, the novel BvHV species were analyzed in their natural host, the bank vole, under experimental conditions, allowing virus propagation and also the collection of first defined diagnostic reference materials.

## 2. Materials and Methods

### 2.1. Bank Voles

Six breeding pairs of bank voles were obtained from a breeding colony at the Federal Environment Agency and used as founder animals of a vole colony at the Friedrich-Loeffler-Institut. The animals were bred and kept under standard animal housing conditions in agreement with European and German animal protection and welfare laws and recommendations. The animals were housed in groups as breeding pairs or individually in polyethylene cages under a 12 h photoperiod at 21 °C and ~60% humidity. Wood shavings, non-bleached pulp and disposable play tunnels were provided as bedding and nesting material. Standard pelleted chow for laboratory rodents (Ssniff, Soest, Germany) and water were available *ad libitum*. Bedding material was changed at least once a week and as required. All animals were inspected visually for the presence of clinical signs; therefore, activity, shortening reaction and social behavior were observed for 5 min per group during bedding change or during daily monitoring. The initially introduced bank voles were tested for pathogens according to FELASA recommendations using C57Bl/6 mice as sentinels [30]. Liver samples from four individuals of the breeding colony (total number of about 400 individuals) tested negative for BvHV by RT-qPCR [25]. One liver sample of a fifth bank vole was analyzed additionally by deep sequencing, and showed no indication of a BvHV infection. For determining the bank vole evolutionary lineages, a PCR targeting a part of the cytochrome *b* gene was performed as previously described [31]. According to this analysis, individuals from the bank vole breeding colony were found to belong to the Western evolutionary lineage.

### 2.2. BvHV Inocula

Free-living bank voles infected with BvHV were identified from animals collected by snap trapping during a monitoring study in Thuringia, southeastern Germany, at a site close to Gotha [32] (strains KS13/914 and KS13/920) and during live trapping of voles at the island of Riems, Mecklenburg-Western Pomerania, northeastern Germany (strains FA07 and F208) (permit numbers: Mecklenburg-Western Pomerania LALLF 7221.3-2.1-030/09; Thuringia TV22-2684-04-04-101/13). Carcasses from snap trapping were stored frozen at −20 °C until dissection was performed according to standard protocols [33,34]. Live-trapped animals from the island of Riems were euthanized under Isofluran^®^ (CP-Pharma Handelsgesellschaft mbH, Burgdorf, Germany) anesthesia and directly dissected. Blood samples were collected during decapitation. Furthermore, to exclude Puumala orthohantavirus (PUUV) infection, blood or visceral cavity lavage of the trapped bank voles was investigated by PUUV-IgG ELISA, as described previously [35].

### 2.3. Experimental Design of Infection Studies

Five infection experiments were performed, four using bank voles and one using the interferon α/β receptor (IFNAR) double knock-out (-/-) C57/Bl6 mice (Figure 1).

All animal experiments were carried out in accordance with the German Animal Welfare Act and approved by the Committee on the Ethics of Animal Experiments of the Federal State of Mecklenburg-Western Pomerania (LALLF M-V/TSD/7221.3-2-028/13). Bank voles of the colony—except the breeding pairs—were housed in groups of equal sex and similar age. Individuals for the experimental studies were selected by breeding success in the colony; the group size also depended on breeding results. Age and sex composition differed among experimental groups (Appendix A). All animals were inspected by handling and visually during the cleaning of the cages. Thus, parameters (activity, posture, shortening reaction, position of the ears, pelage, body weight, nose bulge, circumferential girth tested on palpation) were observed at least once a week. Any modifications of the health assessment are described in the individual experimental section. Animals were classified as “Healthy” or “No clinical signs” if no changes compared to the status on the day before inoculation were seen.

Experiment 1: First, we aimed to learn if colony bank voles could become infected with any of the four wild-type (wt) BvHV strains identified. Four groups of three or four bank voles each were inoculated with material harvested from the BvHV-positive wild-trapped bank voles. To increase our chances of success, BvHV-RNA-positive materials were inoculated via the intraperitoneal (i.p.) route (100 µL), as well as the intramuscular route (i.m.) (30 µL). For all further experiments, only the i.p. route was used. The inoculates consisted of BvHV-positive blood for wt-FA07 (*Hepacivirus F*) and wt-F208 (*Hepacivirus J*), and of visceral cavity lavage fluid for wt-KS13/914 (*Hepacivirus F* and *Hepacivirus J*) and KS13/920 (*Hepacivirus J*). To learn more about infection dynamics of the different strains, single bank voles from each group were euthanized 5, 10, 14 or 28 days post-inoculation (d.p.i.). The exception was the wt virus strain KS13/914 group, which was composed of only three bank voles. In this group, the last time point was omitted. Phosphate-buffered saline (PBS) was inoculated into a fifth group as a control. For virus detection blood, serum and organ samples were collected at the time of euthanasia and stored at −70 °C until further analyses.

Experiment 2: Next, we inoculated in-house bred IFNAR-/- mice with BvHV, to identify potential host species barriers and possible standard models for future studies. This experiment was performed with two of the four original samples: wt-FA07 (*Hepacivirus F*) and wt-F208 (*Hepacivirus J*). Two groups of ten animals each, and one group of nine animals, males and females of variable ages, were inoculated i.p. with 100 µL pooled liver suspension of vole numbers 1-1 to 1-4 from experiment 1 (passage 1 [p1] FA07) or from wild-trapped animals harboring BvHV strains wt-FA07 or wt-F208 (Figure 1, Appendix A). Five animals per group were euthanized on 8 d.p.i. and the five or four remaining animals on 14 d.p.i. For viral RNA detection, blood and liver samples were collected and directly analyzed with hepacivirus RT-qPCR. Animals were selected by breeding success and groups were housed and composed as in the breeding facilities without sexual contact.

Experiment 3: For determining a dose–response effect of viral exposure, five groups composed of three groups of five males and two groups of five female bank voles (Appendix A) were inoculated with variable dilutions of pooled liver suspension of FA07-infected animals 1-1 to 1-4 (p1-FA07, *Hepacivirus F*) or of a wild-trapped vole harboring wt-F208 (*Hepacivirus J*). Each animal was injected i.p. with a volume of 100 µL. Specimens for the inoculation were chosen based on highly positive RT-qPCR results and diluted as shown in Figure 1. For the wt-F208 inoculated voles, one animal per group was euthanized on 8 d.p.i.; the remaining four or five animals were euthanized on 15 d.p.i. None of the p1-FA07 inoculated voles were euthanized on 8 d.p.i., as results from experiment 1 indicated that this was not necessary because the duration of infection was longer for this virus strain than for the wt-F208 virus strain. For virus detection, blood and organ samples were collected at the time of euthanasia and stored at −70 °C until further analyses. For further detection of a dose-related effect, livers from voles inoculated with wt-F208 were sampled in formalin for histopathological analyses. Livers from three uninfected bank voles from the colony were used as controls for this histopathological analysis. For p1-FA07 inoculated bank voles, tissues were not sampled in formalin.

Experiment 4: In order to characterize the long-term course of BvHV infection, one group of four males and one group of four female bank voles were inoculated at the age of ten weeks with the same pooled liver suspension of p1-FA07 (*Hepacivirus F*) from animals 1-1 to 1-4 (Figure 1). All animals were weighed and body temperature was measured daily for the initial 10 d.p.i. After 10 days, body weight was measured once a week and blood samples were taken on 90 d.p.i. One animal per group was euthanized at 90 d.p.i., and the remaining three animals per group were euthanized at 160 d.p.i. Different tissue samples and blood were collected and stored at −70 °C until further analyses (Appendix A). In order to determine if histopathological changes could be found in livers of bank voles inoculated with p1-FA07 (*Hepacivirus F*) inoculated bank voles, livers were similarly scored as livers from wt-F208 (*Hepacivirus J*) inoculated bank voles from experiment 3.

Experiment 5: In order to determine potential transmission routes for BvHV infection in bank voles, two groups—one with males and one with females—were used. From these two groups (male group with four bank voles; female group with five bank voles; all voles 6 months old), two males and three females were arbitrarily selected for inoculation, while the remaining two bank voles each were used as sentinels. Selected animals were inoculated i.p. with 100 µL of pooled liver suspension of p1-FA07 (*Hepacivirus F*) from voles 1-1 to 1-4 (Appendix A). Animals were housed together per group. During this experiment, blood samples were used for biochemical analysis, enabling us to follow liver enzyme activity in blood overtime. For this purpose, one group of five bank voles was housed separately as a negative control group for the biochemical analysis. One blood sample of an animal from the breeding colony was added as additional negative control. All animals were euthanized on 30 d.p.i. and blood as well as liver samples were collected and stored at −70 °C until further analyses by hepacivirus RT-qPCR. EDTA blood was analyzed with VetScan chemistry comprehensive test—mammalian liver profile (Abaxis, Griesheim, Germany). We tested if the enzyme blood value differences between groups were significant by using the Kruskal–Wallis test. We used an alpha level of 0.05 for all statistical tests.

Furthermore, liver samples of the male group were prepared for electron microscopy. For in situ hybridization (ISH), to test both for viral tropism and for the detection of possible routes for virus excretion, haired skin, oral cavity with salivary glands and tongue, stomach, small and large intestines, gall bladder, pancreas, kidney, ureter, urethra, urinary bladder, nasal cavity, lung, liver and male and female genital tract were sampled. Additionally, to test for viral tropism, mesenteric, tracheobronchial and peripheral lymph nodes, spleen, heart, central nervous system and adrenal gland were analyzed by ISH with the FA07 (*Hepacivirus F*)-derived probe.

### 2.4. Sampling Procedure and RNA Extraction

Blood samples were taken with 0.4 × 20 mm Terumo Neolus^®^ cannula (Terumo, Eschborn, Germany) from the facial vein or during autopsy from the heart with an EDTA Multivette 600^®^ (Sarstedt, Nümbrecht, Germany). Samples for RNA analyses were stored at −70 °C and organ samples fixed in 10% (volume/volume, *v/v*) neutral-buffered formalin for ISH. RNA extraction was performed with the KingFisher™ Flex Magnetic Particle Processor (Thermo Fisher Scientific Inc., Waltham, MA, USA) and the Mag Attract Virus Mini M48 Kit (Qiagen, Hilden, Germany), according to the manufacturer’s descriptions.

### 2.5. RT-qPCR Detection of BvHV RNA

All samples were analyzed with RT-qPCR using the qScript XLT One-Step RT-qPCR Tough Kit (Quanta Biosciences, VWR, Hannover, Germany), and strain-specific as well as broad-spectrum hepacivirus primers and probes were used, as previously reported [25] and shown in Table 1. An internal control (IC-2, an EGFP transcript) was added to each sample to control for inhibition of the RT-qPCR reaction.

### 2.6. Virus Genome Sequencing

For the determination of the genomic sequence of original or bank vole passaged BvHV strains (FA07, p1-KS13/920, p1-KS13/914), RNA was extracted from 500 µL BvHV-infected bank vole sera or liver tissue using TRIzol^®^ Reagent (Invitrogen, Darmstadt, Germany) and RNeasy Mini spin columns (Qiagen, Hilden, Germany) as previously described [36]. cDNA synthesis and library preparation were performed as previously described [37]. Sequencing was performed with an Illumina MiSeq instrument (Illumina, San Diego, CA, USA). Raw sequence data were analyzed and mapped using the Genome Sequencer software suite (v. 2.8; Roche, Mannheim, Germany) and Geneious software suite (v. 6.1.6; Biomatters, Auckland, New Zealand). Classification of the BvHV isolates was performed by phylogenetic analyses of complete genome sequences including reference strains MN242371.1 and KC411777.1 [25] using maximum-likelihood methods in MEGA5 [38] (Appendix A).

### 2.7. In Situ Hybridization and Microscopic Evaluation

Liver samples were fixed in 10% (*v*/*v*) neutral-buffered formalin and embedded in paraffin. Samples were collected from 15 animals infected with wt-F208 (*Hepacivirus J*), the control animals from experiment 3 and all animals from experiments 4 and 5 (Appendix A). A 4 μm thick section of each liver sample was routinely stained with hematoxylin and eosin for light microscopic evaluation of liver tissue, including portal triads. In each liver section, 25 portal triads were arbitrarily selected and scored for the presence of lymphocytes and neutrophils. The number of lymphocytes in or directly adjacent to each portal triad was counted. In each liver section, the number of mitotic figures and the number of apoptotic or degenerated cells were counted in each of 25 arbitrarily selected 400×-magnification fields of liver tissue. Scored characteristics are shown in Appendix A. Scoring was performed without the identity of the bank vole being revealed. We tested if counted cells and characteristics differed significantly between groups by using the Kruskal–Wallis test. We used an alpha level of 0.05 for all statistical tests.

In order to determine the cell type tropism over time in the p1-FA07 (*Hepacivirus F*)-infected bank voles, sections of liver and other organs of bank voles were examined for the presence of BvHV RNA by ISH using a specific DNA probe targeting the NS3 gene of wt-FA07 and the RNA scope 2.0 kit (Advanced Cell Diagnostics, San Francisco, CA, USA), as described previously [39]. In brief, 5 μm thick tissue sections were deparaffinized and pre-treated with buffers provided by the manufacturer. Subsequently, the specific probe was hybridized for two hours at 40 °C and the signal was amplified by six amplification steps and visualized with FastRed. Ubiquitin C (UBC)-specific probe was used as positive control. Dihydrodipicolinate reductase (DapB) probe was used as an internal negative control.

### 2.8. Transmission Electron Microscopy

Liver specimens were fixed in glutaraldehyde, post-fixed in osmium tetroxide, embedded in epon and routinely processed for transmission electron microscopy. Ultrathin sections were cut and analyzed in a transmission electron microscope JEM 1400 (Jeol, Tokyo, Japan), as already described [40].

## 3. Results

### 3.1. Collection of BvHV-Positive Animals

During rodent monitoring in Germany [32], bank voles were trapped regularly and investigated for different pathogens [33]. Two bank voles (FA07, F208), trapped in August 2013 in Mecklenburg-Western Pomerania, northeastern Germany, tested positive for BvHV RNA. According to cytochrome *b* sequence analysis, these bank voles belong to the Eastern evolutionary lineage. Another gravid individual (F766) from the same site tested positive for BvHV by RT-qPCR, while all five fetuses were BvHV-RT-qPCR-negative. Additionally, two animals (KS13/914, KS13/920) trapped at a site close to Gotha, Thuringia, southeastern Germany, were found to be BvHV RNA-positive. These two bank voles belonged to the Western evolutionary lineage.

The clade/species identity of the four BvHV strains was analyzed by comparative study using lin1 and lin4 RT-qPCR assays and next-generation sequencing technologies using bank vole samples (Appendix A). These investigations revealed that two inoculates belong to clade 1/*Hepacivirus J* (F208, KS13/920), one to clade 2/*Hepacivirus F* (FA07) and one to both clades/species. Full genome sequence information will become available in the INSDS databases under project PRJEB48702.

### 3.2. Inoculation of Bank Voles with Blood and Visceral Cavity Lavage of Wild-Trapped Bank Voles—Inoculation Experiment 1

In an initial proof-of-principle trial, a total of 15 bank voles from the breeding colony at the FLI (Western lineage) were inoculated with BvHV-positive blood or visceral cavity lavages of the four trapped bank voles FA07, F208, KS13/914 and KS13/920 (Figure 1). Thirteen out of fifteen animals were positive for BvHV RNA on the day of necropsy (5, 10, 14 and 28 d.p.i.) in different organs (Appendix A). In 13 animals, liver and/or blood samples were found to be RT-qPCR-positive (cycle threshold [Ct] value: 18–35). The two RT-qPCR-negative animals had been inoculated with strain F208 (*Hepacivirus J*) and were sacrificed on days 14 and 28 p.i. For the following experiments, blood and liver samples were analyzed by RT-qPCR to determine if hepaciviral infection had occurred.

All inoculated animals remained healthy and physiologically active during the whole experiment, without obvious clinical signs.

### 3.3. Inoculation of IFNAR-/- Mice with Liver Suspension of Bank Voles—Inoculation Experiment 2

In order to evaluate potential host barriers of BvHV, three groups of IFNAR-/- mice were inoculated with original, highly BvHV genome-positive liver suspensions (Ct value: 18) of bank voles FA07 (wt-FA07, *Hepacivirus F*) or F208 (wt-F208, *Hepacivirus J*), or the bank vole passaged FA07 (p1-FA07) material (originating from animals 1-1 to 1-4 of experiment 1) (Figure 1, Appendix A). All 29 animals of this trial remained healthy during the entire experiment. Blood and liver samples of all individuals euthanized at 8 and 14 d.p.i. were negative for BvHV RNA.

### 3.4. Bank Vole Infections with Different Dosages of wt and Bank Vole-Passaged BvHV—Inoculation Experiment 3

Diluted liver suspensions of p1-FA07 (*Hepacivirus F*) and diluted blood suspensions of wt-F208 *Hepacivirus J*) were inoculated in two (dilution 10^−1^ and 10^−2^) and three groups (dilution 10^−1^, 10^−2^, 10^−3^) of bank voles, respectively (Figure 1). Blood samples from all animals inoculated with the two different dilutions of p1-FA07 material were positive for BvHV RNA (Ct values: 22–26) at 15 d.p.i., and 9 out of 10 liver samples of these animals were BvHV-RNA-positive (Ct values: 19–27). No obvious clinical signs were observed.

In contrast, liver samples from only 3 of the 14 animals inoculated with wt-F208 were BvHV-RNA-positive (Ct 35–40), and 12 out of 14 blood samples (Ct 32–38) were RT-qPCR-positive at the end of the experiment (8 or 15 d.p.i.; Appendix A). A single animal of the group inoculated with the highest dose of the wt-F208 material died at 8 d.p.i. without prior clinical signs, and without any grossly visible pathologic changes. The liver of this animal tested positive for BvHV RNA (Ct 31). The four control voles inoculated with liver suspension of a PBS-inoculated control animal of experiment 1 were negative for BvHV RNA, as expected.

Analysis of formalin-fixed paraffin-embedded liver samples from 11 bank voles, inoculated with wt-F208 liver suspension and euthanized at 15 d.p.i., revealed chronic hepatitis with predominantly portal inflammatory infiltrates (both lymphocytes and neutrophils) in all animals (Figure 2a,b and Appendix A), with significant differences between groups.

Lymphocyte infiltration in the portal triads showed a trend to be higher in the group with the highest infection dose compared to the two groups with the lower infection doses (Figure 2b). Additionally, the level of hepatocellular necrosis showed a trend to be higher per virus dose, and no necrotic hepatocytes were seen in the control group (Figure 2a). Neutrophil infiltration (one to four neutrophils per portal triad) could be identified in all infected groups, but not in the control group. There was a trend to higher numbers of neutrophils in voles inoculated with higher virus doses. Neutrophil infiltration could not be identified in the control group voles (Figure 2a). Liver samples from the bank voles inoculated with p1-FA07 strain were not collected for histopathological analysis.

### 3.5. Long-Term Experimental BvHV Infection in Bank Voles—Inoculation Experiment 4

In order to determine the long-term course of BvHV infection, four male and four female bank voles at the age of ten weeks were inoculated i.p. with liver suspension of p1-FA07 (*Hepacivirus F*) of experiment 1 (Figure 1). Although body weight of infected voles alternated over time, none of the animals showed clinical signs. Liver samples of the two animals euthanized at 90 d.p.i. and liver samples of five out of six animals euthanized at 160 d.p.i. were RT-qPCR-positive (Appendix A). The vole 4-7 that was tested negative in the BvHV RT-qPCR at 160 d.p.i. in blood, liver and further organ samples (Appendix A) also tested negative in a blood sample collected at 90 d.p.i., suggesting that inoculation had failed to cause infection in this vole. Liver and blood samples of BvHV-positive animals showed higher RNA levels (Ct 17–24) than the tested lung, heart, brain, spleen, kidney and gall bladder samples (Ct 25–43).

Hepatic lesions in p1-FA07 (*Hepacivirus F*)-infected voles were similar to those in voles infected with the wt-F208 (*Hepacivirus J*) strain, described in experiment 3. Microscopic evaluation further revealed that hepatocellular necrosis was present at 90 and 160 d.p.i., although the frequency was not significantly different (Figure 2c). Furthermore, lymphocytic infiltration was found in the liver of BvHV-RNA-positive animals tested both at 90 and 160 d.p.i., and appeared to be greater at 160 d.p.i. compared to 90 d.p.i. (Figure 2d). All liver samples, hepatocytes, endothelial cells and occasionally vessel lumina stained positive for BvHV RNA with an wt-FA07-specific ISH probe (Figure 3a,b).

Gall bladder did not stain positive. Despite the duration of infection (90 or 180 days), there was no evidence for hepatic fibrosis in the p1-FA07 (*Hepacivirus F*)-infected animals in comparison to the control animals (Figure 3c). No significant lesions were detected in other organs examined (gall bladder, brain, lung, heart, spleen, kidney, pancreas, adrenal gland, stomach, intestine, ovary, uterus, testicle).

### 3.6. Potential Transmission Route of BvHV—Inoculation Experiment 5

Five bank voles were inoculated i.p., at 6 months of age, with pooled p1-FA07 (*Hepacivirus F*) liver material of voles 1-1 to 1-4 from experiment 1, and were kept together with four non-infected bank voles (sentinels) (Figure 1, Appendix A). All animals were euthanized at 30 d.p.i. or days post-contact, and blood and liver were analyzed by RT-qPCR. All five inoculated animals (5-1, 5-4, 5-6, 5-7, 5-9) were highly positive for BvHV-RNA in blood and/or liver samples with Ct values of 19 to 33, whereas all four sentinel animals were negative in blood and liver (Appendix A). All inoculated animals had mild or moderate lymphocytic portal hepatitis and two of four sentinel animals had mild lymphocytic portal hepatitis. Biochemical analysis for the activity of liver enzymes (alanine-aminotransferase, amylase and alkaline phosphatase) in blood samples of the BvHV-inoculated animals of this trial showed no evidence of liver damage in comparison to a group of six negative control animals and the sentinel animals (Figure 4).

To evaluate via which routes—skin, respiratory tract, urinary tract, gastro-intestinal tract, reproductive tract—BvHV might be excreted, the relevant organs from three RT-qPCR-positive animals (5-4, 5-6 and 5-9) and one sentinel animal (5-8) were analyzed with the strain wt-FA07-derived BvHV-specific probe by ISH (Appendix A). All samples except for the liver of the RT-qPCR-positive animals were negative in the ISH. All samples including liver of the sentinel animal were negative in the ISH. These results clearly suggest BvHV in bank voles is exclusively hepatotropic (Figure 3a,b). Based on these results, there was no evidence for excretion via any of the above routes.

### 3.7. Electron Microscopy

Glutaraldehyde-fixed liver samples of four bank voles from experiment 5—two p1-FA07 (*Hepacivirus F*) inoculated, hepacivirus RNA-positive (voles 5-1 and 5-4) and two negative sentinel animals (voles 5-2 and 5-3)—were analyzed by transmission electron microscopy, without prior knowledge of the infection status of the voles. Hepatocytes in the liver samples from both infected animals, but not those from uninfected animals, demonstrated an enlarged endoplasmic reticulum and increased numbers of vesicles as well as lipid droplet accumulation. In addition, in hepatocytes of one infected animal, organelle membranes and lysosomal storage vesicles were detected (Figure 5). Lipid droplet accumulation was confirmed with immunohistochemical stains against the lipid droplet-associated proteins perilipins-2,-3 and -4. This confirmed that lipid accumulation in hepatocytes of bank voles infected with *Hepacivirus F* strain is similar to what has been described for hepatocytes of humans chronically infected with HCV [41].

## 4. Discussion

Here, we describe experimental infection and efficient passaging of two different BvHV species (*Hepacivirus F* and *Hepacivirus J*) in their natural small mammal reservoir host. Experimental infection in bank voles was characterized by liver tropism, with abundant infection of hepatocytes that was consistently associated with lymphocytic hepatitis. Our experimental model managed to induce lesions consistent with earlier observations in naturally infected voles [25], suggesting that our experimental infection managed to mimic a natural hepacivirus infection, and allowed us to study its pathogenesis with knowledge of time, dose and route of inoculation. One strain of BvHV (F208, *Hepacivirus J*) caused an infection that was cleared from the liver after approximately 15 d.p.i. in most voles, while another (FA07, *Hepacivirus F*) caused a persistent infection still abundantly present at 160 d.p.i. We found no evidence for a route of transmission via skin, respiratory tract, urinary tract, gastro-intestinal tract or reproductive tract, and thus, the natural route of transmission of this virus remains unclear.

Typical histopathologic characteristics of HCV-related chronic hepatitis, such as necrosis, mitotic figures, lymphocytic infiltration of portal tracts and neutrophils in the acinar liver parenchyma [42,43], were present in the experimentally infected bank voles, while absent or present in significantly lower frequency in the control animals. Despite the hepatic changes, none of the experimentally infected bank voles showed any clinical signs such as continued weight loss or anorexia. Levels of liver enzymes amylase, alkaline phosphatase as well as alanine aminotransferase of infected bank voles were in the same range as negative control animals. Still, subtle clinical signs might be hard to detect, and liver enzyme levels in the blood were only analyzed at one time post-inoculation, and thus, fluctuations might have easily been missed.

Hepatic lesions were co-localized with viral RNA based on ISH results. Due to the lack of a suitable cell culture system for virus isolation, the inocula for all experiments had to be taken directly from voles, and both BvHV species can only be propagated in its natural host. This makes it harder to be certain that the microscopic lesions in the liver are caused by BvHV infection. However, there was no evidence for parasitic or bacterial co-infections during microscopic examination or through the elicitation of clinical data. Moreover, no further relevant pathogens were detected by next-generation sequencing on selected sera from the BvHV-inoculated bank voles. Furthermore, the changes of the liver parenchyma are similar to what might be expected with hepacivirus infection [8,16].

BvHV infection in bank voles could therefore be a promising small animal model to study the pathogenesis of early stages of hepacivirus infections. Although manifestations such as increased levels of liver enzymes or fibrosis were not seen, HCV-like microscopic hepatic lesions such as lymphocyte infiltration, hepatocellular necrosis and the detection of viral BvHV RNA by ISH were similar to those in equine hepacivirus-infected horses at 10–60 weeks after experimental infection [16]. In addition, electron microscopy analyses of liver samples of two FA07 (*Hepacivirus F*)-infected voles revealed ultrastructural changes—lipid droplet accumulations, enlarged endoplasmic reticulum and increased numbers of vesicles—that were also shown for HCV infection in humans [44]. The progression of HCV infection to the severe stage of liver changes (such as hepatocellular carcinoma) can take up to 144 weeks, as shown for experimentally HCV-infected tree shrews [8], while the life span of bank voles is usually shorter [45]. Furthermore, hepatotoxic factors such as augmented alcohol consumption, drug treatment, co-infection with human immunodeficiency virus (HIV) and old age lead not only to higher rates of liver cirrhosis but also to a faster progression of the disease in humans [46,47,48,49]. Therefore, it is possible that experiment 4 with *Hepacivirus F* strain was still too short to allow the disease to progress to more severe pathological changes characteristic of later stages of HCV infection. For the future, it must therefore be shown whether the observed chronic inflammatory response in the experimentally infected voles will further progress to cirrhosis or even hepatocellular carcinoma during their life span [8].

Interestingly, the BvHV strains FA07 (*Hepacivirus F*) and F208 (*Hepacivirus J*) showed different results in the RNA loads of the liver over time after experimental infection. The strain F208 seemed to be cleared from the liver at 15 d.p.i. in 10 of 11 inoculated bank voles, fitting with a limiting type of infection. For the strain FA07, on the other hand, genome load was found to continue to be present in the liver up to 160 d.p.i. fitting with a chronic persistent pattern of infection. Still, both strains caused inflammation and lymphocytic infiltration. Although livers of the F208 experiment tested negative, surprisingly, 9 of 11 blood samples did test positive for BvHV-RNA at 15 d.p.i. (Appendix A). The origin of this BvHV-RNA in blood is not clear, and it might indicate involvement of other organs than the liver during infection. For interpreting viral clearance or persistence, it would be better to read out by virus culture from these and other samples, rather than reading out by RNA presence. This was not possible because BvHVs, like other hepaciviruses, are very difficult to culture [50,51]. Development of such assays would be of great value in further studies.

Many characteristics of the observed experimental BvHV infections in bank voles have similarities to HCV infections in humans. It was described that some HCV genotypes lead to chronic infection and hepatocellular carcinoma, whereas others induce milder symptoms or can be easily cleared under antiviral treatment [52,53,54,55]. In mice inoculated with Norway rat hepacivirus, viral clearance was shown to be dependent on interferon and CD4+ T helper cells [20]. The results of our experiments 1 and 3 support the possibility of a strain-dependent clearance of the virus in bank voles. The mechanism behind this clearance should be further studied in future experiments. Furthermore, serological tests for the detection of hepacivirus-specific antibodies, such as an immunofluorescence assay or ELISA, are still missing for bank voles, hampering the study of the interplay between the virus and the host’s immune response.

Experiment 3 showed a dose-dependent effect of BvHV wt-F208 (*Hepacivirus J*) inoculation in bank voles, with higher doses associated with more severe inflammatory changes in the liver. HCV-induced cirrhosis and carcinoma are thought to be the effect of the immune system rather than a direct virus effect [56]. Interestingly, a high dose of HCV, as during blood transfusion-acquired HCV infection in humans, leads to a higher risk for primary liver cancer and a higher mortality rate compared to people acquiring HCV infection by sexual transmission, with a presumably lower transmission dose [57,58]. Thus, in HCV, there also seems to be a dose-dependent effect of the virus.

The main transmission route for HCV is parenteral, as by blood transfusion and intravenous drug abuse, but some studies also suggest saliva as a possible origin of infection [59]. The transmission route for BvHV among bank voles is not known. We could show that it is possible to infect animals with BvHV-positive blood, liver suspensions or visceral cavity lavages by parenteral inoculation (i.p. and i.m. or only i.p.). The analyzed intestine and urine samples had only low levels of viral RNA, and the sentinel animals without sexual contact from experiment 5 all remained BvHV-RNA-negative, which suggests that the direct contact transmission of BvHV in bank voles is inefficient, or even not possible, although sexual transmission routes were not studied (sexes were kept separately). Alternatively, a parenteral transmission route, e.g., by arthropod vectors or within-species aggression leading to blood–blood contact, might be responsible for natural infections in free-living bank voles. RT-qPCR-positive results of a gravid vole but negative results of the corresponding fetuses may indicate that a vertical mother–fetus transmission route does not occur in natural BvHV infection [60].

## 5. Conclusions

Bred bank voles can be easily and reproducibly infected by vole liver- or blood-derived bank vole-passaged *Hepacivirus F* and *Hepacivirus J*. The failure of replication of both species in IFNAR-/- mice indicates a species barrier of the bank vole-associated hepaciviruses. The course and the pathological signs of the infections, as well as the suggested route of transmission, seem to be similar to that of HCV infection in humans. Therefore, our experimental in vivo model of BvHV infection in bank voles should be further exploited as a small animal model to understand the host–virus interaction of hepaciviruses, including the clearing of infection and virus transmission. Experimentally infected voles could be used for future development of antiviral intervention strategies and their efficacy testing. The bank vole/BvHV (*Hepacivirus F* and *Hepacivirus J*) model provides a small animal infection model with advantages regarding costs, housing, breeding, group sizes and tools.

## Figures and Tables

**Figure 1 viruses-13-02421-f001:**
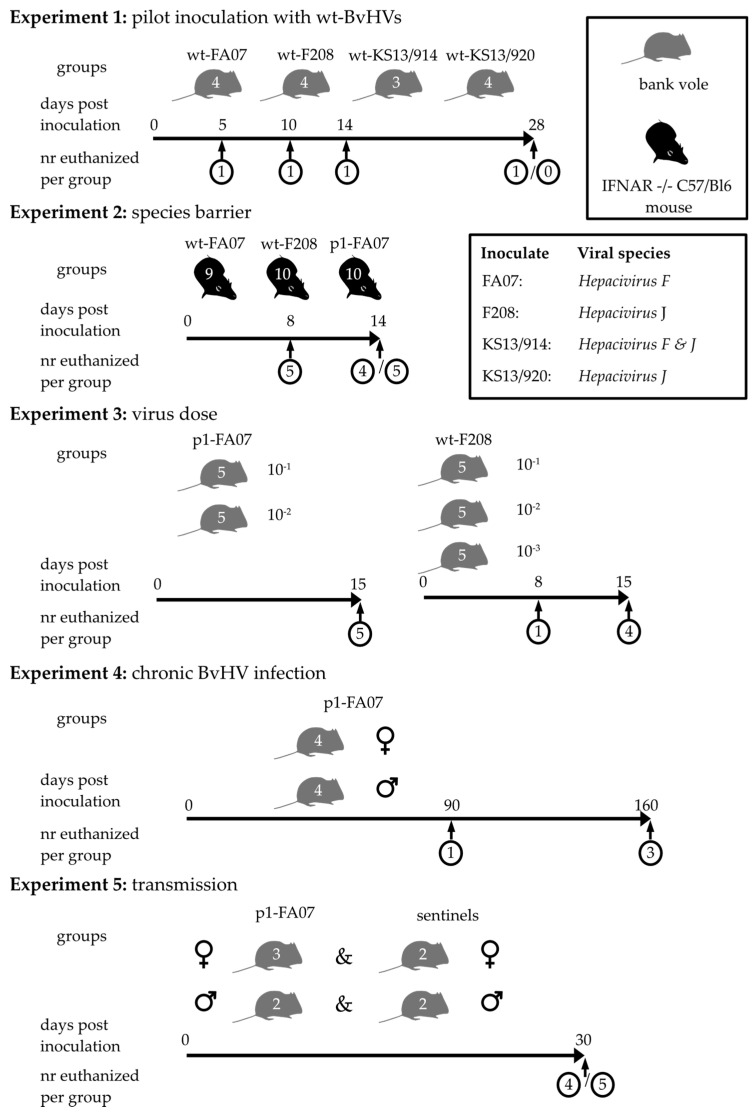
Experimental setup of five bank vole experiments. Four bank voles from the island of Riems, Mecklenburg-Western Pomerania (FA07 and F208), and close to Gotha, Thuringia (KS13/914 and KS13/920), were initially identified to carry BvHVs by RT-qPCR. RT-qPCR-mediated typing and high-throughput sequencing revealed single infection with *Hepacivirus F* (FA07), *Hepacivirus J* (F208 and KS13/920) and a double infection with both BvHVs (KS13/914). Blood or visceral cavity lavage of these animals was used to inoculate bred bank voles intraperitoneally (i.p.) and intramuscularly (i.m.) in experiment 1. Experiment 2 was performed with mice inoculated with liver suspension with wt-FA07, wt-F208 and bank vole passaged FA07 (p1-FA07; a pool of liver suspensions of voles 1-1 to 1-4 from experiment 1). Further experiments were performed with i.p. inoculation of this p1-FA07 or of original wt-F208 strain. Numbers in the vole graph indicate the number of individuals per trial. All control groups were omitted from the graphs.

**Figure 2 viruses-13-02421-f002:**
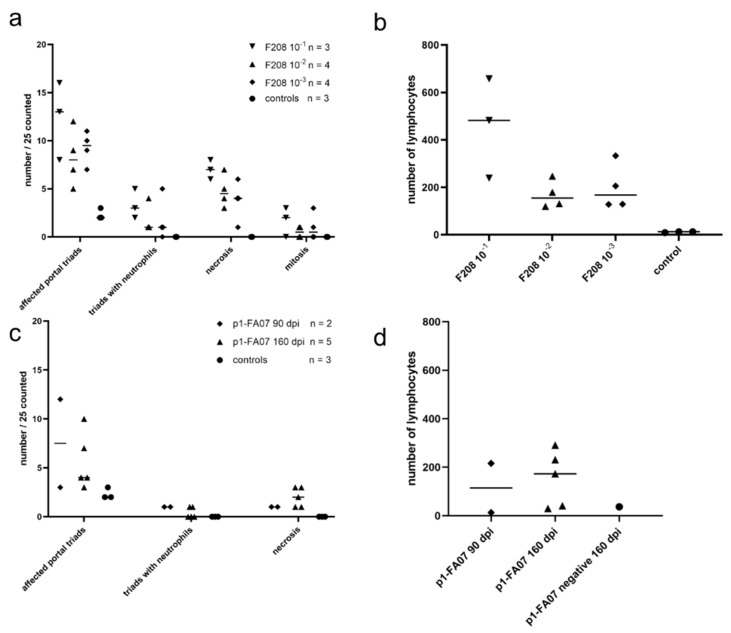
Scatter plots showing individual values and medians (indicated by the horizontal lines) for wt-F208 (*Hepacivirus J*)-inoculated bank voles (experiment 3) and p1-FA07 (*Hepacivirus F*)-inoculated bank voles (experiment 4). Per animal, 25 liver triads were evaluated, and the affected portal triads, triads with neutrophils and lymphocytes were counted. In addition, per animal, 25 arbitrarily selected 400× fields were evaluated for the number of necrotic cells (degeneration) and cells in mitosis. (**a**,**b**) Experiment 3. Analyses of the dose-dependency of the pathological consequences of wt-F208 (*Hepacivirus J*) infection in bank voles. Differences between groups are significant for affected portal triads (*p* = 0.02), for triads with neutrophils (*p* = 0.03), for necrosis (*p* = 0.005) and for number of lymphocytes (*p* = 0.03), but not for mitosis (*p* = 0.390) using the Kruskal–Wallis test. (**c**,**d**) Experiment 4. Pathological consequences of p1-FA07 (*Hepacivirus F*) infection in bank voles at different time points post-inoculation. Statistical testing was not meaningful for these data, because of the low numbers of animals and lesions detected.

**Figure 3 viruses-13-02421-f003:**
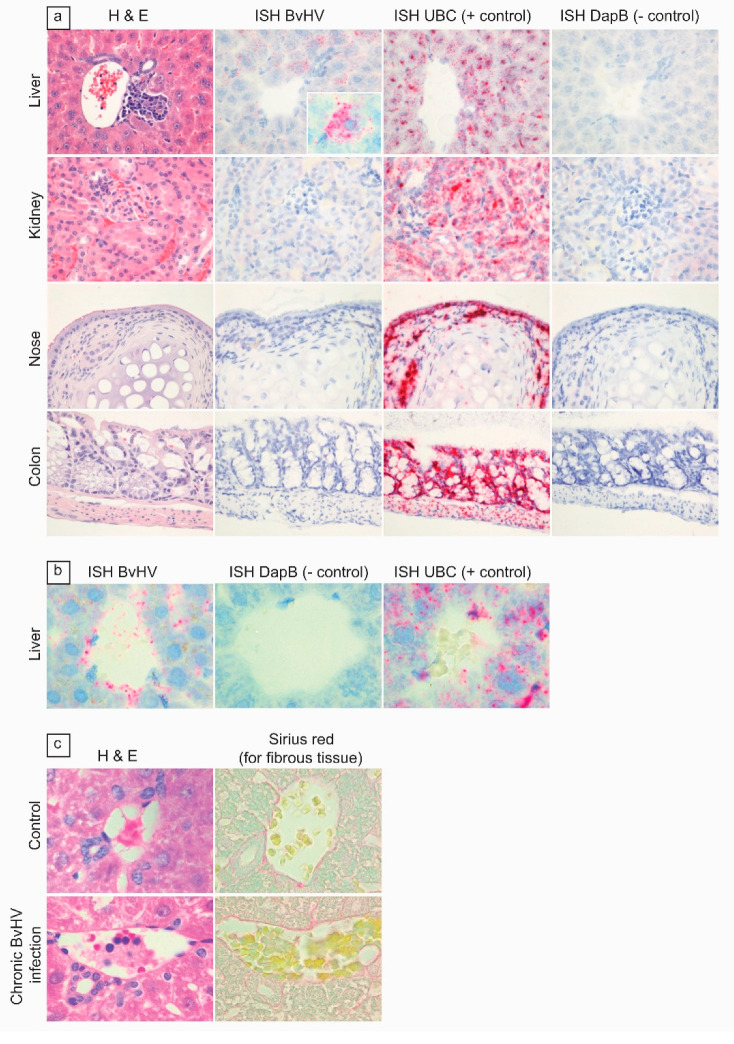
Additional stains for evaluation of cell tropism and fibrosis in BvHV-infected bank voles (p1-FA07, *Hepacivirus F*). (**a**) Rows, from top to bottom: liver (400× magnification), showing lymphocytic portal hepatitis by hematoxylin and eosin (H&E) stain, and associated positive staining of hepatocytes by *Hepacivirus F*-specific ISH (inset: BvHV-specific ISH probe showing hepatocyte with positive staining at higher magnification); kidney (400× magnification), respiratory epithelium of the nose (400× magnification), and colon (200× magnification), showing neither evidence of pathological changes by H&E, nor positive staining by BvHV-specific ISH; Ubiquitin C (UBC)-specific ISH probe, as positive control for ISH; Dihydrodipicolinate reductase (DapB)-specific ISH probe, as negative control for ISH. (**b**) From left to right: liver showing positive staining of endothelial cells lining a blood vessel by BvHV-specific ISH and UBC-specific ISH probe, as positive control for ISH; DapB-specific ISH probe, as negative control for ISH, at 1000× magnification. (**c**) Panels show absence of hepatic fibrosis at 160 d.p.i., based on H&E and Sirius red staining, in BvHV-infected compared to non-infected control bank vole, at 1000× magnification.

**Figure 4 viruses-13-02421-f004:**
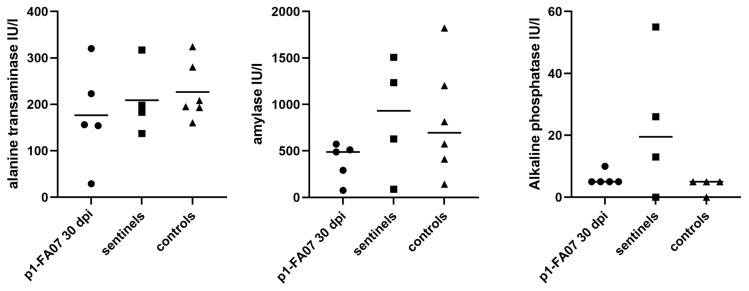
Scatter plots showing individual values and median (indicated by the horizontal black line) of liver enzyme activity levels in blood of p1-FA07 (*Hepacivirus F*)-inoculated voles euthanized at 30 days post-inoculation (indicated by black circles), as well as those of non-infected contact (sentinel) (indicated by black squares) and control (indicated by black triangles) animals, from experiment 5. No significant differences between groups were detected (Kruskal-Wallis, alanine transaminase *p =* 0.6, amylase *p* = 0.2, and alkaline phosphatase *p* = 0.2). Alanine-aminotransferase, amylase and alkaline phosphatase activity levels of five p1-FA07-infected, four sentinel animals and six non-infected bank voles were analyzed in blood samples with VetScan chemistry comprehensive test—mammalian liver profile. Y axis shows enzyme activity in International Units per liter (IU/l).

**Figure 5 viruses-13-02421-f005:**
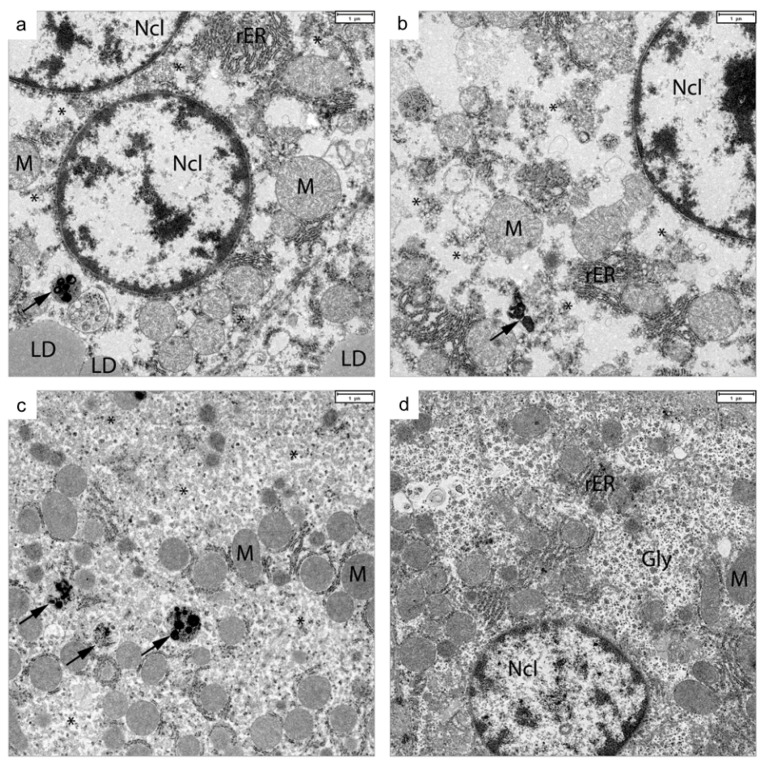
Transmission electron microscopy of liver samples of p1-FA07 (*Hepacivirus F*)-infected animals (**a**–**c**) compared to a liver sample of a control animal (**d**), from experiment 4. Hepatocytes of infected bank voles show vesicular endoplasmic reticulum with organoid membranes (*), lipid droplet (LD) accumulation and pleomorphic lysosomal storage vesicles (arrows). Abbreviations: Gly: glycogen rosettes, LD: lipid droplet, M: mitochondria, Ncl: cell nucleus, rER: rough endoplasmic reticulum. Bar in each panel: 1 µm.

**Table 1 viruses-13-02421-t001:** Forward (F) and reverse (R) primers and fluorescein (FAM) labeled probes used for lin1 and lin4 RT-qPCR detection of *Hepacivirus J* (clade 1) and *Hepacivirus F* (clade 2), respectively.

Designation	Sequence
RHV-NS3-lin1_3.2-F	5′-TGY TGC GAC AGC ACG GCA T
RHV-NS3-lin1_3.3-R	5′-GCG TCC GGR ATT TTR CTC AC
RHV-NS3-lin1_3-FAM	5′-CCG TYG CCT ACT ACC GAG GCG A
RHV-NS3-lin4_4.2-F	5′-ATG ACG GGA TAC ACY GGG AA
RHV-NS3-lin4_4.3-R	5′-CAT KGT SAC TTC ATA TTT GGG CAT
RHV-NS3-lin4_4-FAM	5′-ACT CTG TGT ATG ACA GCT GCY TGA G

## Data Availability

All data are contained within the article or Appendix A.

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
