# Peer review of "The Bank Vole (Clethrionomys glareolus)—Small Animal Model for Hepacivirus Infection"

_viruses, 2021, doi:10.3390/v13122421_

Round 1

Reviewer 1 Report

The authors address all concerns.

Reviewer 2 Report

All of my comments from the first round have been taken into account. Well, from my point of view, it's a really good piece of paper with meaningful content. 

This manuscript is a resubmission of an earlier submission. The following is a list of the peer review reports and author responses from that submission.

Round 1

Reviewer 1 Report

The manuscript by Susanne Röhrs et al. described the pathological changes of bank voles by laboratory infection with bank vole hepacivirus (BvHV), which showed similar liver changes to HCV infection in humans.

My major question about the study was how the virus load was quantified. In line 445-446, the author used ct value as an indication of virus load. I think this is not feasible as no standard curve or internal control was used. Besides, the authors gave no information as to how the tissues were stored, whether they were collected at the same time. This process would also have an effect on the virus viability. So when the author declared that the BvHV strains FA07 and F208 showed different results in the RNA loads of the liver over time after experimental infection, which might also resulted from different virus load as well as virus viability. It is important to make clear in the manuscript how these confounding factors were controlled.

In line 466-468, the authors also pointed out that the pathological changes in laboratory infected bank voles was the same as the naturally infected ones as previous reported. So this study did not make contirbutions to our understanding of this virus.

Author Response

Dear reviewer,

Thank you for the review, and the invitation to submit a revised version of our paper. Hereby, in the text below we provide a list of our responses and a description of the consequent changes we have made in the manuscript. For clarity, the reviewers’ comments are highlighted by making the text bold, while our replies to them are in regular font. The indicated line numbers refer to the numbering in the revised manuscript file with track changes.

Kind regards,

Martin Beer

The manuscript by Susanne Röhrs et al. described the pathological changes of bank voles by laboratory infection with bank vole hepacivirus (BvHV), which showed similar liver changes to HCV infection in humans.

My major question about the study was how the virus load was quantified. In line 445-446, the author used ct value as an indication of virus load. I think this is not feasible as no standard curve or internal control was used.

Reply: Because we did not manage to culture hepaciviruses (see discussion starting line 582 ‘For interpreting viral clearance or persistence it would be better to read-out by virus culture from these and other samples, rather than reading out by RNA presence. This was not possible because BvHV, like other hepaciviruses, is very difficult to culture [48, 49]. Development of such assays would be of great value in further studies.’) Ct values were indeed used as semi-quantitative indication of the respective virus load. We did not correct for efficiency differences of the RT-qPCR test with a standard curve. An internal control (IC-2, an EGFP transcript) was added to each sample and used as an internal control for reverse transcription and PCR amplification (to exclude any inhibitory activity of the sample on the BvHV RT-qPCR). Our intention was to look for virus presence and not for small differences in virus quantity, therefore we have performed this semi-quantitative though precise analysis. We cannot exclude that some of our detected Ct value differences were caused by lack of correction of efficiency differences of the test. However, we expect these differences to be minor, and not relevant for our main conclusions.

We also added now in the materials and methods section, starting line 275 ‘An interntal control (IC-2, an EGFP transcript) was added to each sample to control for inhibition on the RT-qPCR reaction.

Because of the uncertainty raised by the reviewer we have omitted the paragraph ‘Comparative analysis of the different inoculation experiments’, including Figure 6, lines 499 to 520 in the results section, where we had used Ct value differences in a comparison of different organs.

In the result section, the sentence starting in line 364:

In contrast, liver samples from only 3 of the 14 animals inoculated with F208 material were BvHV-RNA-positive with low genome loads (Ct 35 – 40), and 12 out of 14 blood samples (Ct 32 – 38) were RT-qPCR-positive at the end of the experiment (8 or 15 d.p.i.; Table S3).

was changed to:

In contrast, liver samples from only 3 of the 14 animals inoculated with wt-F208 were BvHV-RNA-positive (Ct 35 – 40), and 12 out of 14 blood samples (Ct 32 – 38) were RT-qPCR-positive at the end of the experiment (8 or 15 d.p.i.; Table S3).’

In the discussion section, the sentence starting in line 577

For the strain FA07, on the other hand, genome load was found to increase over time in the liver fitting with a chronic persistent pattern of infection.’

was changed to:

For the strain wt-FA07, on the other hand, genome load was found to continue to be present in the liver up to 160 d.p.i. fitting with a chronic persistent pattern of infection.’

In the results section, one sentence in which Ct values are interpreted then remains in the manuscript, starting in line 419:

Liver and blood samples of BvHV-positive animals showed higher RNA levels (Ct 17 - 24) than the tested lung, heart, brain, spleen, kidney and gall bladder samples (Ct 25 - 43).

In these results the semi-quantitative differences in Ct values are big, even if there was some variation in virus viability between strains, and test efficiency between qPCRs, the measured difference is most likely to be real. The finding confirms the liver tropism of this virus; also this finding fits perfectly with our ISH results. For these reasons we chose to leave it as it is.

Besides, the authors gave no information as to how the tissues were stored, whether they were collected at the same time.

Reply: In material and methods we had described ‘Samples for RNA analyses were stored at -70°C’. To make this information clearer throughout the manuscript we have made the following changes in the matierals and methods section:

For experiment 1 we have changed, starting in line 183:

On 5, 10, 14 and 28 days post inoculation one animal per group was euthanized and blood, serum and organ samples were collected and stored at -70 °C until further analyses.

To:

To learn more regarding infection dynamics of the different strains, four bank voles from each group were euthanized on 5, 10, 14, and 28 days post inoculation (d.p.i.). For virus detection blood, serum and organ samples were collected at the time of euthanasia and stored at -70°C until further analyses.

For experiment 2, we changed, starting in line 206:

Blood and liver samples were collected directly analyzed with RT-qPCR.

To:

For viral RNA detection blood and liver samples were collected and directly analyzed with hepacivirus RT-qPCR.

For experiment 3, we added the following sentence in line 221:

For virus detection, blood and organ samples were collected at the time of euthanasia and stored at -70°C until further analyses.

For experiment 4, we have made the following change starting in line 233:

Different tissue samples and blood were collected for further analyses (Table S4).

To:

Different tissue samples and blood were collected and stored at -70 °C until further analyses (Table S4).

For experiment 5, we have made the following change starting in line 248:

All animals were euthanized on 30 d.p.i. and blood as well as liver samples were analyzed for BvHV by RT-qPCR.

To:

All animals were euthanized on 30 d.p.i. and blood as well as liver samples were collected and stored at -70 °C until further analyses by hepacivirus RT-qPCR.

This process would also have an effect on the virus viability. So when the author declared that the BvHV strains FA07 and F208 showed different results in the RNA loads of the liver over time after experimental infection, which might also resulted from different virus load as well as virus viability. It is important to make clear in the manuscript how these confounding factors were controlled.

Reply: Tissues of voles inoculated with different BvHV strains were stored similarly. We have made this more clear in the materials and methods section (see changes described above).

In line 466-468, the authors also pointed out that the pathological changes in laboratory infected bank voles was the same as the naturally infected ones as previous reported. So this study did not make contributions to our understanding of this virus.

Reply: The comparison to the naturally infected vole is relevant as our model aims to reproduce a natural infection with a hepacivirus. We explained this in the introduction starting in line 75:

Models based on HCV-related Norway rat (Rattus norvegicus) hepacivirus infection or mouse-adapted HCV-strains in mice, knock-out mice, or mice transplanted with human hepatocytes can be inefficient or artificial and do not reflect a naturally occurring virus-host interaction, which is important to study the pathogenesis [8, 10, 19, 20].

Similarity in the lesion induced in our model to the naturally infected bank vole confirms we managed to reproduce that part. To make this clearer we have changed in the discussion our statement in the start of the discussion, strating line 524:

‘Infection in bank voles was characterized by liver tropism, with abundant infection of hepatocytes that was consistently associated with lymphocytic hepatitis, consistent with earlier observations in naturally infected voles [25].’

To:

‘Experimental infection in bank voles was characterized by liver tropism, with abundant infection of hepatocytes that was consistently associated with lymphocytic hepatitis. Our experimental model managed to induce lesions consistent with earlier observations in naturally infected voles [25], suggesting our experimental infection managed to mimic a natural hepacivirus infection and allowed us to characterize its pathogenesis with knowledge of time, dose, and route of inoculation.’

Reviewer 2 Report

For many infectious diseases, good animal models are an important prerequisite for investigating pathomechanisms and developing antiviral drugs and vaccines. The work in the manuscript by Röhrs and co-workers presents an infection model for small animals, which can possibly close this gap for HCV infections. The manuscript describes the implementation and results of five basic animal experiments, which create an important basis for the use of such an infection model.

Even if the here presented results are conclusive, these are sometimes described in the article in a way that is difficult to understand. In particular, Figure 1 should be fundamentally revised here. In addition to errors (9 instead of the 10 animals in group 1 of experiment 2, lack of control animals in experiment 1), the illustration with the legend and the description of the experiments is confusing and difficult to understand. A fixed scheme for the designation of the experiments, groups and the materials used (wt, p1 ...) for inoculation might be helpful here.

Why were different schedules used for experiment 3 for the different groups (no sampling 8 d.p.i. in the animals infected with Bv-FA07)?

The data given in Figure 2 should fundamentally be revised. An one-way ANOVA test basically assumes a normal distribution of the values ​​within a group (which needs tob e tested). This also applies to information on "error bars", the actual content of which is not even explained to the reader at this point. (Here probaly the standard deviation was meant.) However, with the small group sizes shown here, the test for a normal distribution is not permitted. Therefore, from my point of view, a representation of the values ​​in a dot plot (possibly with median) would be expedient here. The Y-axes should also be displayed on the same scale to increase the comparability of the results. In my opinion, statements on the significance of the differences are inadmissible due to the small groups and only change the overall statement slightly. In addition, the authors should double-check the information on the groups in Fig. 2c. According to Table S4 histological samples were obtained only from n = 2 animals at 90 d.p.i. On the other hand, the data from the positive animals at 160 d.p.i. are missing or misslabeled.

From my point of view, dot plots with median would also be more meaningful for Figure 4. My statements made on the significance should also be considered here.

Author Response

Dear reviewer,

Thank you for the review, and the invitation to submit a revised version of our paper. Hereby, in the text below we provide a list of our responses to the comments and a description of the consequent changes we have made in the manuscript. For clarity, the reviewers’ comments are highlighted by making the text bold, while our replies to them are in regular font. The indicated line numbers refer to the numbering in the revised manuscript file with track changes.

Kind regards,

Martin Beer

For many infectious diseases, good animal models are an important prerequisite for investigating pathomechanisms and developing antiviral drugs and vaccines. The work in the manuscript by Röhrs and co-workers presents an infection model for small animals, which can possibly close this gap for HCV infections. The manuscript describes the implementation and results of five basic animal experiments, which create an important basis for the use of such an infection model.

Even if the here presented results are conclusive, these are sometimes described in the article in a way that is difficult to understand. In particular, Figure 1 should be fundamentally revised here. In addition to errors (9 instead of the 10 animals in group 1 of experiment 2, lack of control animals in experiment 1), the illustration with the legend and the description of the experiments is confusing and difficult to understand. A fixed scheme for the designation of the experiments, groups and the materials used (wt, p1 …) for inoculation might be helpful here.

Reply: Figure 1 has been revised according to the suggestions of the reviewer. The errors that the reviewer had identified have been corrected in the new figure 1. Indication to the different strains have been changed as suggested throughout the manuscript and supplementary table files. When material was used from the voles in which the different strains were initially identified, ‘wildtype’ or ‘wt-‘  was added. When material was used from voles that were experimentally infected ‘passage 1’ or ‘p1-‘ was used as prefix.

In the legend of Figure 1, starting line 141, the following text was deleted:

  • IFNAR-/- C57/Bl6’ because already indicated in the figure.
  • pooled bank vole liver suspension from experiment 1(voles 1-1 to 1-4)’ as it is now defined by ‘p1-FA07’
  • For experiment 3, different dilutions of Bv-FA07 or F208 were inoculated to groups of five bank voles each. In experiment 4, the potential long-term consequences of BvHV infection were tested by inoculation of eight voles with Bv-FA07. Finally, the potenial of contact transmission of BvHV was investigated in experiment 5. Two groups, one with two, and one with three voles inoculated i.p. with Bv-FA07 were housed together with two sentinel voles in each group. Five bank voles in a third cage sered as reference for biochimal analysis of blood samples. One blood sample of a bank vole from the breeding colony was added for reference.’ as the subaims of each experiment are already added as text in the figure itself.
  • Additionally, timeline graphs indicated time points of inoculation (indicated with ‘0’) and of euthanasia (numbers indicated the days post inoculation).’ as this is already explained in the figure itself.

The following text was added:

  • ‘All control groups were omitted from the graphs’ as this is described in the text describing each experiment, and the figure space does not allow it.

The text describing the different experiments in the materials and methods section has been changed. The main addition in the text is that we added the sub-aims of each experiment, to make understanding of the methodology easier.

This resulted in the following changes:

line 103: ‘FLI’ was changed to ‘Friedrich Loeffler Institute’

line 117: ...’analyzed additionally by deep desquencing, without any indication’... to ...’analyzed additionally by deep desquencing, and showed no indication’...

line 160: the following text was deleted, for clarity: ‘The bank voles were inoculated with RT-qPCR-positive materials under experimental conditions to allow the in vivo propagation of BvHV and the collection of different types of reference materials.’

line 165: ‘animals’ was replaced by ‘bank voles’ for clarity and potential for confusion with mice

starting line 175: The old text describing experiment 1 was replaced with: ‘First we aimed to learn if colony bank voles could become infected with any of the four wildtype (wt) BvHV strains identified. Four groups of three or four bank voles each were inoculated with material harvested from the wild BvHV-positive wild-trapped bank voles. To increase our chances of success, BvHV-RNA-positive materials were inoculated via the intraperitoneal (i.p.) route (100 µl), as well as the intramuscular route (i.m.) (30 µl). For all further experiments, only the i.p. route was used. The inoculates consisted of BvHV positive blood for wt-FA07 and wt-F208, and of visceral cavity lavage fluid strains wt-KS13/914 and KS13/920. To learn more about infection dynamics of the different strains, single bank voles from each group were euthanized on 5, 10, 14, or 28 days post inoculation (d.p.i.). The exception was the wt-virus strain KS13/914 group, which was composed of only three bank voles. In this group the last time point was omitted. Phosphate-buffered saline (PBS) was inoculated into a fifth group, as a control. For virus detection  blood, serum and organ samples were collected, at the time of euthanasia and stored at -70 °C until further analyses.’

For experiment 2, starting line 197, we added: ‘Next, we inoculated IFNAR-/- mice with BvHV, to identify potential host species barriers and possible standard models for future studies. This experiment was performed with two of the four identified BvHV strains: wt-FA07 and wt-F208. In-house bred mice were used.

For experiment 3, starting line 209 we added: ‘For determining a dose-response effect of viral exposure,’...

For experiment 4, line 225 we added: ‘In order to characterize the long-term course of BvHV infection,’...

and starting line 233: ‘In order to determine if histopathological changes could be found in livers of bank voles inoculated with p1- FA07 inoculated bank voles, livers were similarly scored as livers from wt-F208 inoculated bank voles from experiment 3.

For experiment 5, starting line 236 we added: ‘In order to determine potential transmission routes for BvHV infection in bank voles, two groups—, one  with males, and one with females— were used.’

and in line 245 ‘for the biochemical analysis’

and starting in line 254: ..‘to test both for viral tropism and for the detection of possible routes for virus excretion,’ and line 259 ‘to test for viral tropism’

Why were different schedules used for experiment 3 for the different groups (no sampling 8 d.p.i. in the animals infected with Bv-FA07)?

Reply: From experiment 1, shown in Table S1, we had a suggestion that wt-F208 caused a limited duration of infection, while wt-FA07 caused a more chronic infection. For wt-F208 inoculated voles, viral RNA could be detected in voles sacrificed on 5 and 10 dpi, but not in voles sacrificed on 14 and 28 dpi. In contrast, for wt-FA07, viral RNA could be detected in the voles on all these time points. For this reason, 8 dpi was added to read out if infection had occurred for p1-F208 inoculated groups, but not to p1-FA07 groups.

To make this more clear in the manuscript, we added in the materials and methods section in the part describing experiment 3, starting in line 216, the following text:

‘None of the p1-FA07 inoculated voles were euthanized on 8 d.p.i. as results from experiment 1 had indicated that this was not necessary, because the duration of infection was longer for this virus strain than for the F208 virus strain.’

The data given in Figure 2 should fundamentally be revised. An one-way ANOVA test basically assumes a normal distribution of the values ​​within a group (which needs tob e tested). This also applies to information on "error bars", the actual content of which is not even explained to the reader at this point. (Here probaly the standard deviation was meant.) However, with the small group sizes shown here, the test for a normal distribution is not permitted. Therefore, from my point of view, a representation of the values ​​in a dot plot (possibly with median) would be expedient here.

The Y-axes should also be displayed on the same scale to increase the comparability of the results.

Reply: Figure 2 was changed as suggested to a dot plot. We changed the Y-axes as suggested by the reviewer.

In my opinion, statements on the significance of the differences are inadmissible due to the small groups and only change the overall statement slightly.

Reply: To be conservative, we used the Kruskal-Wallis test (which does not require the assumption of normal distribution of values) instead of the one-way ANOVA for testing the significance of the differences between the groups. This test showed that, despite the small group sizes, the number of portal triads affected, the triads with neutrophils and presence of necrosis, as well as the total numbers of lymphocytes were significantly different between the different groups. We agree with the reviewer to be careful to interpret the data with such small groups, and we did not think further testing for statistically significant differences was meaningful. There was a trend for an increasing number of lesions in the higher dose group compared to lower and middle dose groups, and controls.

We have changed figure 2 as suggested, and changed the text regarding these results accordingly.

In the materials and methods section regarding the microscopic evaluation, starting line 301, we added the statistical test used and the chosen alpha level:

We tested if counted cells and characteristics differed significantly between groups by using the Kruskal-Wallis test. We used an alpha level of 0.05 for all statistical tests.’

and starting line 250, regarding blood enzyme levels, from voles in experiment 5:

We tested if the enzyme blood value differences between groups was significant by using the Kruskal-Wallis test. We used an alpha level of 0.05 for all statistical tests.’

In the results section, starting in line 371, we added the underlined text:

Analysis of formalin-fixed paraffin-embedded liver samples from 11 bank voles, inoculated with wt-F208-liver suspension and euthanized at 15 d.p.i., revealed chronic hepatitis with predominantly portal inflammatory infiltrates (both lymphocytes and neutrophils) in all animals (Figure 2a and b, Figure S1 a-c), with significant differences between groups.’

In the continuing line we changed:

Lymphocyte infiltration in the portal triads was significantly higher in the group with the highest infection dose compared to the two groups with the lower infection dose (Figure 2b). Additionally, the level of hepatocellular necrosis was significantly higher per virus dose, and no necrotic hepatocytes were seen in the control group (Figure 2a).

To:

Lymphocyte infiltration in the portal triads showed a trend to be higher in the group with the highest infection dose compared to the two groups with the lower infection doses (Figure 2b). Additionally, the level of hepatocellular necrosis showed a trend to be higher per virus dose, and no necrotic hepatocytes were seen in the control group (Figure 2a).

Figure 2 legend has been changed to:

Scatter plots showing individual values and medians for wt-F208 inoculated bank voles (experiment 3), and p1-FA07 inoculated bank voles (experiment 4). Per animal 25 liver triads were evaluated, and the affected portal triads, triads with neutrophils, and lymphocytes were counted. In addition, per animal, 25 arbitrarily selected 400x fields were evaluated for the number of necrotic cells (degeneration) and cells in mitosis. (a, b) Experiment 3. Analyses of the dose-dependency of the pathological consequences of wt-F208 infection in bank voles. Differences between groups are significant for affected portal triads (p = 0.02), for triads with neutrophils (p = 0.03), for necrosis (p = 0.005), and for number of lymphocytes (p = 0.03), but not for mitosis (p = 0.390) using the Kruskal-Wallis test. (c, d) Experiment 4. Pathological consequences of p1-FA07 infection in bank voles at different time points post inoculation. Statistical testing was not meaningful for these data, because of the low numbers of animals and lesions detected.’

In addition, the authors should double-check the information on the groups in Fig. 2c. According to Table S4 histological samples were obtained only from n = 2 animals at 90 d.p.i. On the other hand, the data from the positive animals at 160 d.p.i. are missing or misslabeled.

Reply: Indeed, we had mislabeled the figure. We have corrected this in our new figure.

From my point of view, dot plots with median would also be more meaningful for Figure 4. My statements made on the significance should also be considered here.

Reply: We agree and modified figure 4 and the legend according to these suggestions.

Reviewer 3 Report

The authors described that bank vole hepacivirus species (clade 1 and 2) could infect bank voles but not IFNR(-/-) B6 mice, and also mentioned that BvHV indicated hepatotropism and persistent or acute infection in dependent on viral strain. This study would be very interesting and the animal models may contribute to development of HCV vaccine. However, there are several issues to be addressed virologically. Since blood or visceral cavity lavage of the naturally infected animals, which are crude samples, were employed as inoculates in this study, the accurate effect of BvHV may not be elucidated from the data shown in Fig 4 and 5. In addition, anti-HCV agents should be tested for confirmation of BvHV infection, because the previous reports indicated that some anti-HCV agents (LNA-miR122, sofosbuvir) could suppress Norway rat hepacivirus infection. Suppression of BvHV infection by treatment with an anti-HCV agent may help to confirm BvHV infection-dependent effect. It would be better for the authors of this manuscript to consider experimental designs again.

Minor

  • BvHV clade 1 and 2 may be classified into hepacivirus J and F, respectively, on the basis of the reports of Drexler et al (PLoS Pathog 2013) and ICTV classification. It seems better to described it in the text.
  • The authors should quantify time-dependent propagation of virus (e.g., copy/ng RNA) in blood or tissue samples prepared form the infected bank voles.